# Characterization of Differential GPX4 Essentiality Between Intrahepatic and Extrahepatic Cholangiocarcinoma via Leveraging of a Large-Scale Functional Genomic Screen

**DOI:** 10.3390/ijms262411990

**Published:** 2025-12-12

**Authors:** Ye Rim Lee, Chaeyoung Seo, Md Abdullah, Su Hyun Baek, Seung Jin Lee

**Affiliations:** 1AI-Biology & Pharmaceutical Science, Chungnam National University, Daejeon 34134, Republic of Koreaellalux@naver.com (C.S.); 2College of Pharmacy, Chungnam National University, Daejeon 34134, Republic of Korea; abdulpharm1985@gmail.com (M.A.); bsh6922@naver.com (S.H.B.)

**Keywords:** cholangiocarcinoma, enzyme-inhibited ferroptosis, glutathione peroxidase 4, WNT, DepMap, bioinformatics

## Abstract

Ferroptosis has emerged as a promising therapeutic vulnerability of diverse malignancies, yet the regulatory circuits adopted by each in cholangiocarcinoma (CCA) subtypes remain incompletely understood. We integrated the genome-wide CRISPR–Cas9 loss-of-function screens and transcriptomic profiles of the Cancer Dependency Map and then systematically assessed the essentiality of ferroptosis suppressor genes (FSGs) in the intrahepatic (iCCA) and extrahepatic (eCCA) subtypes. Nineteen and 16 essential FSGs were identified in iCCA and eCCA, respectively, among which GPX4 exhibited a significantly higher dependency in iCCA. Pharmacological inhibition of GPX4 with RSL3 markedly reduced cell viability and induced lipid peroxidation in iCCA cell lines, whereas eCCA cell lines displayed pronounced resistance associated with elevated GPX4 expression. A transcriptomic comparison revealed enrichment of WNT signaling in eCCA. Co-treatment with the tankyrase inhibitor XAV-939 and RSL3 enhanced growth inhibition of eCCA cells, indicating that WNT signaling contributed to ferroptosis resistance. These findings indicate that iCCA exhibits a preferential dependency on GPX4, whereas WNT–β-catenin signaling mediates resistance in eCCA. Collectively, the results clarify the molecular basis of subtype-specific ferroptosis vulnerability and offer a rationale for combinatorial therapeutic strategies that integrate GPX4 and WNT pathway inhibition when treating refractory eCCA.

## 1. Introduction

Ferroptosis has emerged as a promising therapeutic vulnerability across diverse malignancies [1,2]. This iron-dependent cell death is driven by the peroxidation of polyunsaturated fatty acid–containing phospholipids (PUFA-PLs), which are intrinsically susceptible to oxidative damage [1,2]. Pro-ferroptotic processes involve the generation of PUFA-CoAs by acyl-CoA synthetase long-chain family member 4 (ACSL4), their incorporation into phospholipids by lysophosphatidylcholine acyltransferase 3 (LPCAT3), and the oxidation of PUFA-PLs to lipid hydroperoxides (PUFA-PL-OOH) by Fe^2+^ and lipoxygenases (LOXs) [1,2].

To counteract these pro-ferroptotic processes, cells deploy multiple ferroptosis defense systems that either detoxify lipid peroxides or reduce the pool of oxidizable PUFA-PLs. The central detoxifying axis is mediated by glutathione peroxidase 4 (GPX4), which reduces lipid hydroperoxides using glutathione synthesized from intracellular cysteine supplied by system x_c_^−^ [2,3]. Several GPX4-independent antioxidant pathways further suppress lipid peroxidation. Ferroptosis suppressor protein 1 (FSP1; also known as apoptosis-inducing factor mitochondria-associated 2, AIFM2) acts as an NADH-dependent oxidoreductase that regenerates ubiquinol (CoQ_10_H_2_) and vitamin K hydroquinone (VKH_2_) [2,3]. Dihydroorotate dehydrogenase (DHODH) contributes to the regeneration of ubiquinol within the inner mitochondrial membrane [2,3]. The guanosine triphosphate cyclohydrolase 1 (GCH1) pathway produces tetrahydrobiopterin (BH_4_), a potent radical-trapping antioxidant, while dihydrofolate reductase (DHFR) regenerates BH_4_ from dihydrobiopterin (BH_2_) using NADP^+^/NADPH [2,3]. On the other hand, lipid-remodeling enzymes restrict ferroptosis by decreasing PUFA-PL availability. Stearoyl-CoA desaturase 1 (SCD1) converts saturated fatty acids into monounsaturated fatty acids (MUFAs), which are resistant to peroxidation [2,3]. Membrane-bound O-acyltransferase domain-containing 1 and 2 (MBOAT1 and MBOAT2) selectively incorporate MUFAs into lysophosphatidylethanolamine, thereby increasing MUFA-containing PE species [4]. Microsomal glutathione S-transferase 1 (MGST1) interacts with arachidonate 5-lipoxygenase (ALOX5) and epigenetically suppresses ALOX15 activity, limiting lipid peroxidation [5]. Together, these ferroptosis defense axes—including GPX4/system x_c_^−^/glutathione, FSP1/CoQ_10_H_2_, DHODH/CoQ_10_H_2_, GCH1/DHFR/BH_4_, SCD1/MUFA, MBOAT1,2/MUFA, and MGST1/ALOXs—form a multilayered network. However, their relative contributions appear highly context-dependent, and the specific regulatory circuitry governing ferroptosis resistance in each tumor subtype remains incompletely understood.

Cholangiocarcinoma (CCA) is relatively uncommon, but remains the second most prevalent primary liver cancer after hepatocellular carcinoma, characterized by a poor prognosis and 5-year survival rates below 20% [6,7]. Anatomically, CCA is classified into intrahepatic (iCCA), perihilar (pCCA), and distal (dCCA) subtypes according to the site of origin. Historically, the latter two subtypes were collectively termed extrahepatic CCA (eCCA), but the updated 2021 classification now distinguishes them as pCCA and dCCA. [6,7]. In Western populations, pCCA is the most common subtype (50–60%), followed by dCCA (20–30%) and iCCA (10–20%) [6,7]. Despite the overlapping histopathological features, CCA subtypes exhibit distinct molecular architectures and therapeutic tractability. iCCA hosts emerging targetable changes including IDH1/2 mutations and FGFR2 fusions, and frequent mutations in chromatin-remodeling genes (BAP1, ARID1A) [6]. In contrast, PRKACA and PRKACB fusions as well as mutations in ELF3 and ERBB3 are more frequently observed in eCCA. KRAS, TP53, and SMAD4 mutations occur across all CCA subtypes but are modestly more frequent in large-duct-type iCCA and in eCCA [6]. Microsatellite instability/mismatch repair deficiency occurs in only 1–2% of CCA [7]. Approximately 30–40% of patients with iCCA harbour potentially actionable molecular alterations, whereas the proportion is substantially lower in eCCA [6,7]. These subtype-dependent molecular features are also likely to influence the regulatory mechanisms of ferroptosis. For example, mutant IDH1/2, which is frequently observed in iCCA, consumes NADPH and diminishes endogenous GSH-regeneration capacity, thereby increasing susceptibility to lipid peroxidation [8]. In contrast, ERBB2/3 and ELF3 mutations [9,10], which are enriched in eCCA, as well as KRAS and TP53 mutations [1,2,6,7], which occur across all subtypes, have been reported to suppress ferroptosis by inducing SLC7A11 and GPX4 expression or disrupting redox balance, ultimately promoting tumor progression.

Targeting of the ferroptosis defense machinery has recently gained increasing attention in CCA. Among the major ferroptosis defense mechanisms, GPX4 is the most extensively studied component and represents a central regulator of CCA progression. Elevated GPX4 expression correlates with reduced overall survival and drives metabolic reprogramming toward enhanced glucose utilization [11]. Disrupting the multiple oncogenic circuits that sustain aberrant GPX4 upregulation has been shown to promote ferroptosis. These circuits include STAMBPL1-, PAX8-AS1-, and METTL3-mediated NRF2 activation [12,13,14]; SHARPIN-mediated p53 induction [15]; and JUND/linc00976-mediated miR-3202 suppression [16]. Additional resistance mechanisms involving aberrant expression of SLC7A11 and FSP1, as well as NCOA4-mediated ferritinophagy and altered metabolism, have been reported [17,18,19]. However, these studies were conducted almost exclusively in iCCA models, using cell lines and organoids. Although several studies used both iCCA and eCCA cell lines to identify ferroptosis regulators such as AKR1C1, ACSL3/4, FADS2, and CNNM4 [20,21,22,23,24], they relied on only one or two models and did not examine their relevance to subtype-specific molecular features. Given the distinct molecular features of CCA subtypes, a systematic investigation encompassing both iCCA and eCCA is needed to determine whether ferroptosis susceptibility would differ according to subtype-specific biology.

Leveraging information in large cancer cell line datasets allows systematic evaluation of gene function and therapeutic vulnerability, in turn offering mechanistic insights into context-specific target identification [25]. The Cancer Dependency Map (DepMap) project systematically profiles gene essentiality across more than 1000 human cancer cell lines through genome-wide CRISPR–Cas9 loss-of-function screens [26,27]. The resulting dependency score quantifies the impact of each gene knockout on cell proliferation, normalized to the reference distributions of non-essential and pan-essential genes. Integrative analyses that combine DepMap dependency data with genomic and transcriptomic information have revealed the synthetic lethal interactions or subtype-specific targetable vulnerabilities of multiple tumor types [26]. In this study, we integrated large-scale gene dependency scores and expression datasets from DepMap and identified the essential ferroptosis suppressor genes (FSGs) of iCCA and eCCA. We also found that GPX4 is more essential in iCCA than in eCCA. We experimentally validated this differential dependency and further investigated the strategies to overcome the ferroptosis resistance of eCCA. These findings reveal the molecular basis of ferroptosis resistance and highlight potential, combinatorial therapeutic strategies for eCCA.

## 2. Results

### 2.1. Prioritization of Essential FSGs in iCCA and eCCA and Selective GPX4 Vulnerability

Gene dependency scores derived from CRISPR–Cas9 loss-of-function screens in the DepMap dataset quantify the impact of gene knockout on cell viability, reflecting the extent to which individual genes are required for survival [26,27]. Gene essentiality is quantified using a normalized dependency score based on non-essential and pan-essential reference sets [27,28]. A score of 0 indicates no effect, whereas 1 corresponds to the median effect of universally essential genes; genes with scores >0.5 are generally considered essential [27,28]. The dependency dataset comprised 22 iCCA and 6 eCCA cell lines, and the transcriptome dataset included 36 CCA cell lines; thus, the 28 cell lines with both CRISPR dependency profiles and matched RNA-sequencing data were selected for analysis (Figure 1A). To ensure robust estimation of gene essentiality, we stratified these 28 models into iCCA and eCCA and applied two criteria: (i) an absolute dependency-score cutoff and (ii) statistically significant differences between subgroups.

To identify essential FSGs based on the absolute dependency-score criterion, we applied a series of filtering steps. First, genes not expressed in more than 5% of CCA models were excluded because sufficient transcriptional activity is a prerequisite for meaningful interpretation of dependency scores. Second, genes lacking dependency scores in over 20% of the 28 models were removed to minimize artifacts associated with data sparsity [20]. After these steps, dependency scores for 11,618 genes across 28 cell lines were retained. Subsequently, genes with median dependency scores > 0.5 were classified as essential. Finally, genes showing essentiality in fewer than 10% of CCA cell lines were excluded as low-generalizability targets, and genes annotated as “common essential” in DepMap were removed as broadly essential genes across normal cells [28]. These processes identified 268 essential genes in iCCA and 295 in eCCA, which were subsequently intersected with the curated FerroDb v3 gene list to define essential ferroptosis suppressor genes (FSGs).

For each essential FSG, we calculated the median dependency score across iCCA or eCCA cell lines and the fraction of cell lines in which the dependency score exceeded 0.5. These metrics were visualized as two-dimensional plots (Figure 1B). Among these, 19 genes in iCCA and 16 in eCCA overlapped with known FSGs (Figure 1B and Table 1). A set of nine commonly essential FSGs shared by both subtypes—including SLC3A2, FXN, SKP2, RICTOR, and AHCY—were directly or indirectly linked to the cystine/System x_c_^−^/GSH/GPX4 axis, one of the major ferroptosis defense pathways [1,29,30]. iCCA cell models exhibited increased dependence on GPX4 within this axis. Other FSGs showing higher essentiality in iCCA, such as UBIAD1, PTPMT1, CHMP5, SETD2, SDHD, FECH, and ARF6, may directly or indirectly influence iron metabolism, mitochondrial reactive oxygen species (ROS) generation, or membrane integrity [31,32,33]. In eCCA, essential FSGs including KLF5, FOXM1, IGF1R, TEAD1, and SREBF1 regulate the expression of various ferroptosis defense genes, indirectly affecting mitochondrial ROS generation and lipid metabolic processes [1,34].

Next, we further evaluated subtype selectivity among the essential FSGs by assessing both the magnitude and statistical significance of dependency differences between 22 iCCA and 6 eCCA cell lines. Dependency scores were compared using the Wilcoxon rank-sum test, given the non-normal distribution of the data and the sample-size imbalance. To quantify the direction and magnitude of subtype-selective effects, we calculated Hedges’ g, which provides a standardized effect size corrected for small-sample bias [35]. The results were visualized in a volcano plot, with Hedges’ g on the x-axis and −log_10_(*p*-value) on the y-axis (Figure 1C). This identified four differentially essential FSGs: CHMP5 and GPX4 were selectively essential in iCCA cell lines, whereas PARL and ZMYND8 were selectively essential in eCCA cell lines.

Collectively, while the cystine/System x_c_^−^/GSH/GPX4 axis may be essential in both iCCA and eCCA, iCCA appeared to exhibit greater vulnerability in the core ferroptosis defense pathways involving GPX4, UBIAD1, and PTPMT1. Given that GPX4 exhibited a statistically significant differential essentiality between the two subtypes and it can be targeted using available small-molecule inhibitors, we further validated this potential subtype-selective dependency using pharmacological GPX4 inhibition with RSL3.

### 2.2. iCCA-Specific Vulnerability to GPX4 Inhibition

We evaluated the antiproliferative effects of RSL3 in three iCCA cell models (HuCCT1, SSP25, and YSCCC) and three eCCA cell models (TFK1, SNU-245, and SNU-1196) using the CellTiter-Glo^®^ assay. Exposure to RSL3 for three days induced pronounced growth inhibition of iCCA cells, whereas eCCA cells displayed markedly reduced sensitivity (Figure 2A,B). HuCCT1 cells showed the greatest sensitivity to RSL3 (IC_50_ = 0.05 µM), whereas TFK1 cells were the most resistant (IC_50_ = 1.42 µM). In contrast, the response to gemcitabine and doxorubicin were comparable between subtypes, indicating that the observed RSL3 sensitivity was not attributable to a subtype-dependent difference in overall drug response (Figure 2C). The growth inhibition induced by RSL3 in HuCCT1 and SSP25 cells was almost completely rescued by pretreatment with 20 µM ferrostatin-1 for 1 h, but not by necrostatin-1s or Z-VAD-FMK, confirming that ferroptotic cell death was the primary mechanism of action (Figure 2D, left and middle panels). In contrast, in TFK1 cells, growth inhibition by 3 µM RSL3 was partially reversed by ferrostatin-1 (Figure 2D, right panel), suggesting that higher doses of RSL3 may trigger cell death mechanisms beyond ferroptosis.

We next compared the sensitivities to RSL3-induced lipid peroxidation of HuCCT1 and TFK1 cells. Treatment with 1 µM RSL3 for 3 significantly increased lipid peroxidation in HuCCT1 cells, but only a minimal 1.1-fold change was observed in TFK1 cells, as measured by the BODIPY^®^ C11 probe (Figure 3A). Prolonged exposure to RSL3 for 14 h increased lipid peroxidation at concentrations as low as 0.05 µM in HuCCT1 cells, whereas a comparable effect was observed at 4 µM in TFK1 cells (Figure 3B). The RSL3-induced lipid peroxidation of HuCCT1 cells was markedly suppressed by co-treatment with 100 µM deferoxamine or 20 µM ferrostatin-1, confirming that this was an iron-dependent process (Figure 3C). Next, to evaluate the cellular defense capacity against exogenous oxidative stress, we exposed the cells to cumene hydroperoxide (CHP)—a relatively stable organic hydroperoxide that induces secondary lipid peroxidation (Figure 3D). As expected, HuCCT1 cells were susceptible to 30 µM CHP, whereas TFK1 cells remained resistant up to 100 µM and showed a marked response only at 300 µM.

To elucidate the differential sensitivities of these cell models to RSL3, we assessed the basal protein levels of GPX4, as the cellular response to a small-molecule inhibitor often depends on the abundance of its target protein. Measuring GPX4 protein levels is more informative than mRNA levels because GPX4 is a selenoprotein whose translation efficiency is tightly regulated by selenium availability [1,2,3]. The expression of GPX4 was approximately 4.5-fold higher in TFK1 cells than in HuCCT1 cells (Figure 3E). Induction of GPX4 expression addition of selenium to the culture medium effectively prevented RSL3-induced lipid peroxidation (Figure 3F). Collectively, these findings indicate that HuCCT1 cells are highly susceptible to RSL3-induced ferroptosis, whereas TFK1 cells, with elevated GPX4 expression, are resistant to GPX4 inhibition.

### 2.3. Differential Transcriptomic Analyses and Combined Inhibition of WNT Signaling and GPX4 in eCCA

As iCCA cells of the DepMap dataset were more dependent on GPX4 than were eCCA cells (Figure 1C and Table 1), and eCCA cell lines exhibited greater resistance to RSL3 than iCCA cell lines (Figure 2A), we sought strategies to enhance GPX4 inhibition in eCCA models. Transcriptomic profiling of three iCCA and three eCCA cell models identified 531 and 441 differentially expressed genes (DEGs) upregulated in iCCA and eCCA, respectively (Figure 4A). Figure 4B shows the over-representation analysis (ORA) of DEGs revealed functional characteristics that distinguish the iCCA and eCCA groups. Pathways related to *Pertussis*, *Malaria*, and *Focal adhesion* were significantly enriched in the iCCA group (adjusted *p* < 0.05), whereas genes upregulated in eCCA were nominally enriched in pathways associated with *Breast cancer, Alanine, aspartate* and *glutamate metabolism*, and *WNT signaling* (nominal *p* < 0.05, adjusted *p* > 0.05). KEGG pathway mapping further revealed up-regulation of the WNT–AXIN–TCF/LEF axis and down-regulation of JNK signaling in eCCA, suggesting the distinct transcriptional signatures between the subtypes (Figure 4C).

Next, we investigated whether inhibition of canonical and non-canonical WNT signaling could potentiate the antiproliferative effect of GPX4 inhibition by RSL3. TFK1 cells were exposed to 12.5 μM XAV-939, 12.5 μM Y39983, or 12.5 μM SP600125 for 1 h, and further incubated with 1 or 10 μM RSL3 for 72 h. Growth of TFK1 cells was not affected by treatment with 1 µM RSL3 alone; however, co-treatment with 10 µM XAV-939 markedly potentiated growth inhibition to a level comparable to that observed with 10 µM RSL3 alone (Figure 4D). XAV-939 is a tankyrase inhibitor that stabilizes the β-catenin destruction complex, thereby suppressing WNT signaling, indicating that tankyrase inhibition effectively enhances the antiproliferative response to GPX4 inhibition. In contrast, Y-39983, a ROCK1 inhibitor that targets the non-canonical WNT pathway, exerted only a minimal effect on the cellular response to RSL3. SP600125, a JNK inhibitor, alone reduced cell viability by approximately 25% at 3 µM and tended to attenuate the effect of 10 µM RSL3. We further examined whether combining XAV-939 with RSL3 would enhance ferroptosis by assessing lipid peroxidation. Consistent with the viability data, co-treatment with 1 µM RSL3 and 10 µM XAV-939 produced a significant increase in lipid peroxidation compared with RSL3 alone (Figure 4E). These results indicate that the WNT signaling axis plays a critical role in conferring ferroptosis resistance in eCCA cells.

## 3. Discussion

In contrast to other malignancies, molecular studies in CCA rarely utilize diverse cell lines or patient-derived samples and are further constrained by the limited availability of large-scale public datasets. Currently, only 70–80 CCA cell lines have been established, and most available models were generated in East Asia—particularly in China, Japan, Thailand, and Korea—where the disease burden is highest. Notably, eCCA-derived cell lines are approximately threefold fewer than iCCA-derived ones [36]. This imbalance, despite the higher clinical prevalence of eCCA, has contributed to a research landscape dominated by iCCA-based studies. Consistent with this limitation, large-scale DepMap screening datasets include 22 iCCA but only 6 eCCA cell lines; nevertheless, this dataset remains the most extensive resource for identifying potential therapeutic targets. Here, we systematically analyzed these public datasets using robust statistical approaches followed by experimental validation. After stratifying the datasets into iCCA and eCCA, we identified essential FSGs using an absolute dependency-score cutoff as well as statistical comparisons between subgroups. Through integrated bioinformatic and functional analyses, we identified subtype-dependent ferroptosis vulnerabilities, differential sensitivity to GPX4 inhibition, and an effective combinatorial approach involving GPX4 and β-catenin inhibition in eCCA.

Using the absolute-cutoff criterion, we prioritized nine FSGs essential in both subtypes, ten specifically essential in iCCA, and seven in eCCA. The FSGs commonly essential in all CCA cell lines contribute to key ferroptosis-defense processes. ADAR1/2 edits RNA by converting adenosine to inosine and enhances GPX4 and SCD1 expression [37]. SLC3A2 forms a heterodimer with SLC7A11 to constitute the system x_c_^–^ transporter responsible for cystine uptake and glutathione synthesis [1,2]. The methyltransferase ZC3H13 promotes m^6^A modification of peroxiredoxin 6, thereby upregulating SLC7A11 [38], whereas S-adenosylhomocysteine hydrolase (AHCY) regulates the methionine cycle and indirectly supports cysteine and GSH biosynthesis [29]. SKP2 promotes ubiquitination of SLC3A2, NCOA4, and ACSL4, thereby regulating cystine availability, labile iron, and PUFA–CoA pools, respectively [39]. Frataxin (FXN), a mitochondrial iron chaperone, facilitates Fe–S cluster assembly [40], while thymidine phosphorylase (TYMP) and RICTOR regulate PI3K/Akt/mTOR signaling to suppress ferroptosis [41]. Among these candidates, the roles for ACSL4 and m^6^A modifications in ferroptosis regulation have been demonstrated in CCA cell models [21,42], but additional experimental validation across subtypes is still required. This absolute-cutoff criterion is intuitive and aligns with the DepMap consortium’s definition of essential genes; however, essentiality across all CCA cell lines does not guarantee CCA selectivity, as some genes may also be essential in other cell types.

Among the subtype-specific essential FSGs, GPX4, UBIAD1, PTPMT1, CHMP5, SETD2, SDHD, FECH, and ARF6 showed higher mean dependency scores in iCCA, whereas KLF5, FOXM1, IGF1R, PARL, and SREBF1 were preferentially essential in eCCA. We further filtered subtype-specific essential FSGs by Wilcoxon rank-sum test and Hedges’ g, and revealed that CHMP5 and GPX4 were differentially essential in iCCA. CHMP5, an ESCRT-III component, regulates endosomal maturation and lysosomal degradation of membrane proteins and protects against ferroptotic cell death [32]. Notably, GPX4, a druggable master regulator of ferroptosis suppression, exhibited selective essentiality in iCCA, which we independently validated experimentally. Meanwhile, PARL and ZMYND8 showed higher mean dependency scores in eCC than iCCA. PARL, a mitochondrial rhomboid protease, modulates PINK1-dependent mitophagy, maintains respiratory complex III activity, and facilitates the transport of coenzyme Q from mitochondria to the plasma membrane [43]. The histone reader ZMYND8 enhances NRF2 protein stability through KEAP1 silencing [44]. Although CHMP5, PARL and ZMYND8 are not directly druggable, their subtype-specific essentiality suggests context-dependent functions that warrant further investigation.

We attempted to determine whether the mutations commonly identified in clinical CCA samples—IDH1/2, BAP1, ARID1A, ELF3, and ERBB3—were associated with differential GPX4 essentiality in our dataset. However, the number of mutant cell lines was too small to allow meaningful statistical evaluation. Additionally, we assessed whether the expression of key components of the ferroptosis defense axis—including AIFM2, SCD1, and DHODH—could account for subtype-specific differences in GPX4 dependency. None of these genes showed a statistically significant association with GPX4 dependency. Further studies with larger cohorts or integrated multi-omics datasets will be required to delineate subtype-dependent covariates that may govern selective FSG dependency.

In our study, experimental validation of subtype-specific GPX4 dependency in HuCCT1 and TFK1 cells revealed that TFK1 cells were highly resistant to lipid peroxidation induced by either RSL3 or CHP. GPX4 protein expression was markedly higher in TFK1 cells than in HuCCT1 cells, which may explain the enhanced ability to tolerate GPX4 inhibition and mitigate lipid radical accumulation. To identify molecular strategies that could enhance ferroptosis in eCCA, we analyzed differentially expressed genes and found marginal enrichment of several oncogenic pathways, including breast cancer and WNT signaling. Consistent with bioinformatic results, the WNT pathway inhibitor XAV-939 markedly potentiated RSL3-induced growth suppression and lipid peroxidation in TFK1 cells. As the β-catenin/TCF4 transcriptional complex binds directly to the GPX4 promoter to activate expression thereof and thereby suppress ferroptotic cell death [45], these findings imply that tankyrase-mediated β-catenin stabilization may maintain ferroptosis resistance. Given that the WNT–β-catenin pathway is known to be activated in most CCAs [6], the combinatorial effect of WNT inhibition with RSL3 warrants further investigation in additional CCA models. In contrast, combining RSL3 with various signaling inhibitors (e.g., PI3K, Akt, and mTOR inhibitors) did not further enhance the antiproliferative effects of GPX4 inhibition in eCCA cell models in our preliminary studies.

This study has several limitations that warrant consideration. First, the current clinical classification of CCA distinguishes iCCA, pCCA, and dCCA subtypes, but cell lines in DepMap were annotated as iCCA or eCCA. Moreover, iCCA comprises heterogeneous subgroups—large-duct, small-duct, and cholangiolocellular types—with the large-duct subtype sharing molecular characteristics with pCCA and dCCA. Consequently, the available iCCA versus eCCA categorization of cell lines does not fully align with clinically relevant classifications, potentially obscuring subtype-specific biological features. Second, gene dependency in DepMap is derived from CRISPR-based in vitro screening, which captures survival-associated vulnerabilities in monoculture systems. However, CCA tumors are characterized by prominent stromal components and diverse cellular compositions that are not reflected in standard CRISPR screens. Additionally, survival-based dependency metrics may overlook clinically relevant targets that are non-essential for cell survival: for example, antibody–drug conjugate targets are often non-essential surface proteins, and epigenetic regulators may be more vulnerable to pharmacologic inhibition of enzymatic activity than to gene knockout. Finally, the limited number of available cell lines constrains the extent to which our bioinformatic and experimental analyses can capture the clinical heterogeneity of CCA. Despite this limitation, the DepMap dataset remains the most extensive resource for uncovering potential therapeutic targets. The subtype-selective candidates identified in our analyses provide an initial rationale for future studies incorporating larger cohorts, multi-omics integration, and in vivo validation to more definitively characterize subtype-dependent ferroptosis vulnerabilities.

In summary, we prioritized FSGs as potential subtype-specific targets in iCCA and eCCA cell models. To our knowledge, this study provides the first integrative analysis of CCA cell lines incorporating DepMap genome-wide loss-of-function and transcriptomic data. These in silico FSG candidates may serve as potential therapeutic targets for CCA, providing a basis for future mechanistic validation and preclinical assessment in both in vitro and in vivo models. We found that GPX4 inhibition was more effective in iCCA models, and pharmacological blockade of WNT signaling might effectively sensitize eCCA cells to GPX4 inhibition and overcome ferroptosis resistance. Although our findings provide new insights, the study is limited by the mismatch between clinical CCA subtypes and available cell-line annotations, the constraints of survival-based CRISPR screening, and the small number of CCA models. These limitations may be addressed in future studies using larger cohorts and multi-omics integration. Collectively, our integrative approach provides a framework for identifying clinically relevant, subtype-specific vulnerabilities and highlights potential therapeutic strategies for certain CCA subtypes with unmet clinical needs.

## 4. Materials and Methods

### 4.1. Curation of the DepMap Dataset and Prioritization of Essential FSGs

The data used to derive the CERES dependency probability score (hereafter dependency score) (CRISPRGeneDependency.csv, version 25Q2), and to identify common essential genes (CRISPRInferredCommonEssentials.csv, version 25Q2), and measure gene expression (OmicsExpressionRawReadCountHumanAllGenesStranded.csv, version 23Q4; OmicsExpressionProteinCodingGenesTPMLogp1.csv, version 23Q4) were obtained from DepMap (https://depmap.org/portal/, accessed on 13 October 2025). Metadata files (Model_v2.csv, version 24Q4) were used to map Omics Profile IDs to specific cell lines. The dependency score quantifies the likelihood that gene loss affects cell viability, corrected for cell line-specific copy number alterations. The score ranges from 0 to 1, with higher values indicating a greater probability of essentiality in terms of survival. Gene lists for ferroptosis suppressors were retrieved from the FerroDb V3 database (http://www.zhounan.org/ferrdb, accessed on 13 October 2025).

### 4.2. Statistical Analysis of Subtype-Selective Essential FSGs

Group-wise comparisons of dependency scores between the 22 iCCA and 6 eCCA cell lines were performed using the Wilcoxon rank-sum test, which is appropriate for data with non-normal distributions and unequal group sizes. Genes with *p*-values < 0.05 were considered significant. To quantify the magnitude and direction of subtype-selective differences, Hedges’ g was calculated. This yields a standardized effect size that corrects for small-sample bias [35]. The statistics were defined as follows:$$g=J\times \frac{{\bar{x}}_{i}\;-\;{\bar{x}}_{e}}{s_{p}}$$
where ${\bar{x}}_{i}$ and ${\bar{\;x}}_{e}$ denote the mean dependency scores of the iCCA and eCCA groups, respectively, and $S_{p}$ is the pooled standard deviation calculated as:$$s_{p}=\sqrt{\frac{{(n}_{i}\;-\;1)s_{i}^{2}+\;{(n}_{e}\;-\;1)s_{e}^{2}}{n_{i}\;+\;n_{e\;}-\;2}}$$
with $s_{i}$ and $s_{e}$ representing the standard deviations and $n_{i}$ and $n_{e}$ denoting the corresponding sample sizes of the iCCA and eCCA groups. The small-sample correction factor J was applied to obtain an unbiased standardized mean difference and is given by:$$J=\;1-\;\frac{3}{4(n_{i}\;+\;n_{e})\;-\;9}$$

A positive Hedges’ g denotes greater essentiality in the iCCA group, whereas a negative value greater dependency in eCCA. All analyses were implemented in Python (version 3.9; pandas 2.2, pingouin 0.5, matplotlib 3.9), thus ensuring full reproducibility.

### 4.3. Cell Culture and Cell Viability Measurement

HuCCT1, SSP-25, YSCCC, and TFK-1 cells were purchased from the RIKEN BioResource Research Center (Tsukuba, Ibaraki, Japan). SNU-245 and SNU-1196 cells were obtained from the Korea Cell Line Bank (Seoul, Republic of Korea). All cell lines were maintained in either Dulbecco’s Modified Eagle’s Medium (DMEM) (Thermo Fisher Scientific, Waltham, MA, USA) or RPMI-1640 medium (Thermo Fisher Scientific). All cell lines were routinely tested and confirmed to be free of mycoplasma contamination.

Cells were seeded into 96-well plates at 2000–3000 cells per well in fresh culture medium. After 24 h, cells were treated with RSL3 (MedChemExpress, Mon-mouth Junction, NJ, USA), gemcitabine (Selleckchem, Houston, TX, USA), or doxorubicin (Merck KGaA, Darmstadt, Germany) in a dose-dependent manner, or with the vehicle control (dimethyl sulfoxide; Merck KGaA). Ferrostatin-1 (Merck KGaA), Necrostatin-1s (Selleckchem), Z-VAD-FMK (Selleckchem), Z-VAD (Merck KGaA), β-mercaptoethanol (Merck KGaA), XAV-939 (MedChemExpress), Y39983 (Selleckchem), or SP6000125 (Merck KGaA) was added 1 h before RSL3 exposure. Cell viability was assessed at 72 h after treatment using the ATP-based CellTiter-Glo^®^ luminescence assay (Promega, Madison, WI, USA), a widely validated method known for its high sensitivity and reproducibility in drug-response studies. Luminescence was recorded using a Synergy plate reader (BioTek, Agilent Technologies, Santa Clara, CA, USA), following the manufacturer’s instructions. Areas under the curve (AUC) were calculated with the aid of GraphPad Prism software version 10.5.0 (GraphPad Software, Boston, MA, USA).

### 4.4. Monitoring of Lipid Peroxidation

Lipid peroxidation was detected using the Image-iT™ Lipid Peroxidation Kit with the lipophilic BODIPY^®^ 581/591 C11 probe (Thermo Fisher Scientific), which reflects lipid ROS levels through an oxidation-dependent fluorescence shift. Cells were treated with RSL3 or CHP as indicated, and then exposed to the BODIPY^®^ C11 probe for a further 30 min at 37 °C. Cells were collected via trypsinization, washed with PBS, and fluorescence was detected using a Guava^®^ easyCyte flow cytometer (Merck KGaA) with excitation/emission at 488/525 nm; the results were analyzed using InCyte2.6 software (Merck KGaA).

### 4.5. Protein Sampling and Immunoblotting

Whole-cell lysates were prepared using standard lysis buffer and protein concentrations were determined with the Bradford protein assay (Bio-Rad, Hercules, CA, USA), as previously described [46]. Equal amounts of protein (15 μg per sample) were subjected to SDS–polyacrylamide gel electrophoresis on 8 or 12% polyacrylamide gels and transferred onto nitrocellulose membranes that were then incubated with primary antibodies for overnight followed by horseradish peroxidase (HRP)-conjugated secondary antibodies. Protein bands were visualized using an HRP chemiluminescent substrate (Thermo Fisher Scientific) and detected by the iBright CL1000 Imaging System (Thermo Fisher Scientific). The primary antibodies used included anti-GPX4 (#AB125066, Abcam, Cambridge, UK) and anti-β-actin (#SC-47778, Santa Cruz Biotechnology, Dallas, TX, USA). HRP-conjugated secondary antibodies were obtained from Jackson ImmunoResearch (West Grove, PA, USA). Antibodies were used at dilutions from 1:2000 to 1:10,000.

### 4.6. Identification of DEGs and Pathway Enrichment Analysis

To identify enriched pathways in iCCA and eCCA cells, expression normalization and differential expression analyses employed the DESeq2 package of R (version 4. 2.2) with the raw read counts to ensure robust normalization and statistical modeling. DEGs were defined by a |log_2_ fold change| > 1 and an adjusted *p* < 0.05 and visualized using the EnhancedVolcano package (version 1.14.0). ORA was performed using GSEApy (v1.1.10) based on differentially expressed gene sets between iCCA and eCCA. The ORA parameters were min_size = 5, max_size = 2000, cutoff = 0.05. Significant pathways were defined using an FDR < 0.05 or a nominal *p* < 0.05. The Pathview R package (ver.1.48) was used to visualize significantly altered pathways. KEGG pathway-based data integration was applied to map expression values onto pathway diagrams, highlighting up- and down-regulated genes between iCCA and eCCA.

### 4.7. Statistical Analyses

All data are presented as mean ± SD, with n indicating the number of independent experiments. Statistical significance was defined as *p* < 0.05. Depending on data distribution and the type of comparison type, statistical analyses were performed using one-way ANOVA with Tukey’s post hoc test for multi-group comparisons and Wilcoxon rank-sum test for two-group comparisons. Wilcoxon rank-sum test or one-way ANOVA followed by Tukey’s post-hoc test. All analyses were conducted using SPSS Statistics for Windows (version 26; SPSS Inc., Chicago, IL, USA), R (R Foundation for Statistical Computing, Vienna, Austria), or Python (version 3.9; Python Software Foundation), with the appropriate packages indicated.

## Figures and Tables

**Figure 1 ijms-26-11990-f001:**
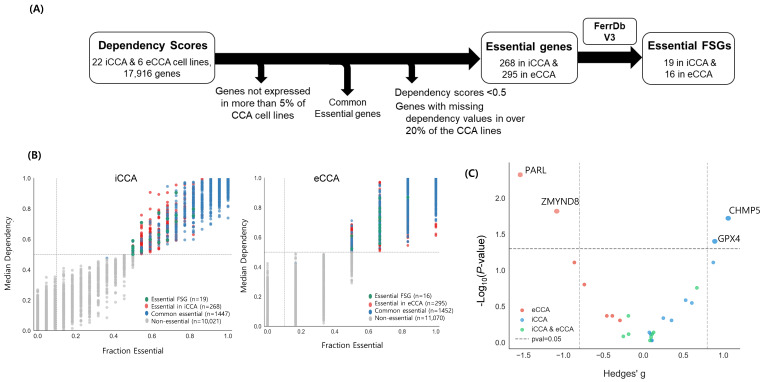
Selectively essential FSGs in iCCA and eCCA using the dependency probability score dataset. (**A**) Gene selection schema. (**B**) Scatter plot of essential FSGs in iCCA (left) and eCCA (right) cell lines. Blue circles represent common essential genes from DepMap, grey circles indicate genes with essentiality < 10% or a median dependency score < 0.5, red circles indicate selectively essential genes, and green circles represent selectively essential FSGs. (**C**) Volcano plot with the Hedges’ g on the x-axis and −log_10_(*p*-value) on the y-axis, highlighting genes with ∣g∣ ≥ 0.8 and *p* < 0. 05. A positive Hedges’ g denotes greater essentiality in the iCCA group, reflected by higher gene dependency scores compared with the eCCA group. Red, blue, and green dots indicate genes preferentially essential in eCCA, iCCA, and both subtypes, respectively.

**Figure 2 ijms-26-11990-f002:**
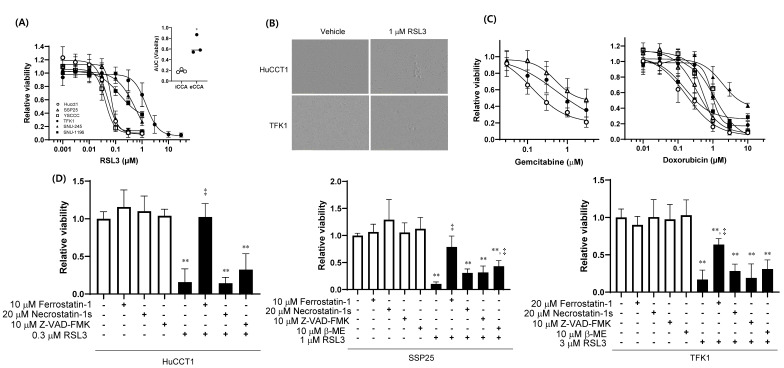
Differential sensitivity of iCCA and eCCA cell lines to RSL3-induced ferroptosis. (**A**) Cells were treated with increasing concentrations of RSL3 for 3 days, and viability was measured using the CellTiter-Glo^®^ assay (n = 4–5 in triplicates). The inset shows the areas under the curve (AUC) for viability within the concentration range of 0.001–1 µM. (**B**) Representative cell images taken at 48 h after 1 μM RSL3 treatment. (**C**) Gemcitabine- (left) and doxorubicin- (right) induced changes in cells over 3 days. Cell line symbols are the same as in (**A**). (**D**) HuCCT1 (left), SSP25 (middle), and TFK1 (right) cells were pretreated with ferrostatin-1, necrostatin-1s, Z-VAD-FMK, or β-mercaptoethanol for 1 h and further exposed to RSL3 for 3 days. Data are presented as mean ± standard deviations (SD) from 3–4 independent experiments, each performed in triplicate. Statistical significance was determined by one-way ANOVA followed by Tukey’s post hoc test. * *p* < 0.05; ** *p* < 0.01 vs. vehicle; ^‡^ *p* < 0.01 vs. RSL3 alone.

**Figure 3 ijms-26-11990-f003:**
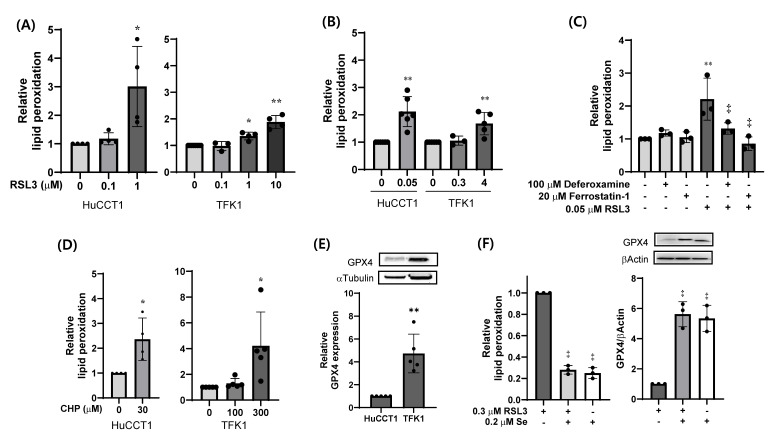
Differential sensitivity of iCCA and eCCA cell lines to RSL3-induced lipid peroxidation. (**A**) HuCCT1 (left) and TFK1 (right) cells were treated with 0.1–10 μM RSL3 for 3 h (n = 4). (**B**) HuCCT1 and TFK1 cells were treated with 0.05–4 μM RSL3 for 14 h (n = 4). (**C**) HuCCT1 cells were pre-treated with 100 μM deferoxamine or 20 μM ferrostatin-1 for 1 h before RSL3 exposure for 14 h (n = 3). (**D**) HuCCT1 (left) and TFK1 (right) cells were treated with 30–300 µM CHP for 2 h (n = 4). (**E**) Basal GPX4 expression in exponentially growing HuCCT1 and TFK1 cells (n = 4). (**F**) SSP25 cells were cultured with or without 200 nM selenium supplementation for 24 h, followed by exposure to 0.3 µM RSL3 for 14 h (n = 3). For panels (**A**–**D**) and ((**F**) left), lipid peroxidation was assessed by flow cytometry with a BODIPY^®^ C11 probe. For panels (**E**) and ((**F**) right), GPX4 expressions were analyzed by Western blotting and quantified relative to either α-tubulin or β-actin. Data represent mean ± SD from independent experiments, as indicated. Statistical significance was assessed using one-way ANOVA with Tukey’s post hoc test for multi-group comparisons and Wilcoxon rank-sum test for two-group comparisons. * *p* < 0.05; ** *p* < 0.01 vs. vehicle; ^‡^ *p* < 0.01 vs. RSL3 alone.

**Figure 4 ijms-26-11990-f004:**
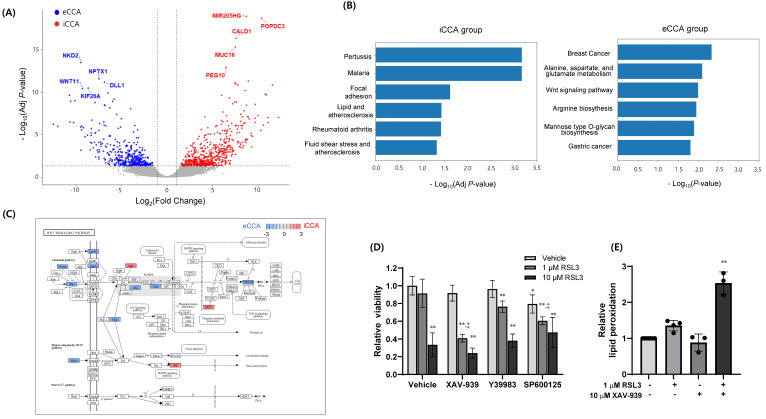
Differential Transcriptomic Analyses and Combined Inhibition of WNT Signaling and GPX4. (**A**) Volcano plot DEGs between three ICC and three ECC cell lines; red and blue dots indicate genes upregulated in iCCA and eCCA, respectively. (**B**) ORA with DEGs in the iCCA (left) and eCCA (right) groups. Significantly enriched pathways were identified in the iCCA group (adjusted *p* < 0.05), whereas genes upregulated in eCCA were only nominally enriched (nominal *p* < 0.05, adjusted *p* > 0.05) and thus shown based on unadjusted significance. (**C**) The KEGG pathway mapping of the WNT signaling pathway, highlighting genes upregulated in iCCA (red) and eCCA (blue). (**D**) TFK1 cells were pretreated with 12.5 μM XAV-939 (a tankyrase inhibitor), 12.5 μM Y39983 (a ROCK inhibitor), or 12.5 μM SP600125 (a JNK inhibitor) for 1 h, followed by exposure to 1 or 10 μM RSL3 for 72 h. Cell viability was measured using the CellTiter-Glo^®^ assay (n = 4–5). (**E**) TFK1 cells were treated with 1 μM RSL3 with or without XAV-939 for 3 h, and lipid peroxidation levels were quantified using the BODIPY^®^ C11 probe (n = 3). Independent experiments were performed as indicated, and all cell viability assays were conducted in triplicate. Data represent mean ± SD and statistical analyses were performed using one-way ANOVA with Tukey’s post hoc test. * *p* < 0.05; ** *p* < 0.01 vs. vehicle alone; ^‡^ *p* < 0.01 vs. RSL3 alone (without inhibitor).

**Table 1 ijms-26-11990-t001:** Median gene dependency scores and fractions of essential cells for FSGs identified as essential in iCCA and/or eCCA cell models.

Essential in Subtype		FSGs	iCCA		eCCA	
			Median Dependency ^1^	Fraction Essential ^2^	Median Dependency ^1^	Fraction Essential ^2^
iCCA & eCCA	1	MED1	**0.9**	0.73	**0.73**	0.67
	2	SLC3A2	**0.8**	0.82	**0.68**	0.67
	3	FXN	**0.75**	0.82	**0.87**	0.83
	4	SKP2	**0.71**	0.72	**0.77**	0.67
	5	TYMS	**0.71**	0.55	**0.86**	0.67
	6	RICTOR	**0.63**	0.64	**0.56**	0.67
	7	ADAR	**0.6**	0.59	**0.58**	0.67
	8	ZC3H13	**0.58**	0.64	**0.61**	0.5
	9	AHCY	**0.54**	0.5	**0.64**	0.67
iCCA	1	GPX4	**0.77**	0.64	0.32	0.16
	2	CHMP5	**0.76**	0.73	0.11	0.33
	3	UBIAD1	**0.73**	0.68	0.38	0.5
	4	PTPMT1	**0.65**	0.68	0.49	0.5
	5	SETD2	**0.59**	0.68	0.46	0.33
	6	SDHD	**0.59**	0.5	0.17	0.16
	7	UBR5	**0.55**	0.5	0.14	0.16
	8	METTL17	**0.52**	0.5	0.4	0.33
	9	FECH	**0.52**	0.55	0.4	0.33
	10	ARF6	**0.5**	0.5	0.25	0.16
eCCA	1	KLF5	0.47	0.45	**0.8**	0.67
	2	FOXM1	0.36	0.18	**0.74**	0.67
	3	IGF1R	0.22	0.31	**0.71**	0.67
	4	TEAD1	0.39	0.45	**0.67**	0.67
	5	PARL	0.26	0.09	**0.64**	0.67
	6	ZMYND8	0.1	0.18	**0.61**	0.67
	7	SREBF1	0.16	0.31	**0.52**	0.5

^1^ Median Dependency: Median value of gene dependency scores in iCCA or eCCA cell lines. ^2^ Fraction Essential: Fraction of cell lines in which the dependency score of a given gene exceeds 0.5. **Bold** indicates Median Dependency > 0.5.

## Data Availability

The datasets presented in this study can be found in DepMap (https://depmap.org/portal/, accessed on 13 October 2025) and the FerroDb V3 database (http://www.zhounan.org/ferrdb, accessed on 13 October 2025). All other data supporting the findings of this study are available within the article.

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
