# Peer review of "Characterization of Differential GPX4 Essentiality Between Intrahepatic and Extrahepatic Cholangiocarcinoma via Leveraging of a Large-Scale Functional Genomic Screen"

_ijms, 2025, doi:10.3390/ijms262411990_

Round 1

Reviewer 1 Report

Comments and Suggestions for Authors

Based on the review of the document, the following suggestions are made to the author:
1. Include more details on the specific mechanisms of ferroptosis and how these relate to CCA subtypes to further strengthen the rationale for the research.
2. Provide more details on the sample size and inclusion/exclusion criteria to fully assess the robustness of the design.
3.    It may be useful to consider including multivariate regression to control for possible confounding variables and gain a deeper understanding of the relationships between variables.
4.    Detailed information on the experimental protocols could be provided, including validation of the methods used and justification for the choice of analytical tools.
5.    The discussion could address the limitations of the study and suggest future directions for research, which would strengthen the discussion and validity of the conclusions.

Thank you.

Author Response

Reviewer #1

Comment 1. Include more details on the specific mechanisms of ferroptosis and how these relate to CCA subtypes to strengthen the rationale.

Response:

We thank the reviewer for this valuable comment. To address it, we have substantially revised the Introduction section to strengthen the mechanistic rationale. Specifically, we now (i) provide an expanded overview of ferroptosis regulation, including pro-ferroptotic processes (1st paragraph) and the major defense systems that detoxify lipid peroxides or limit the availability of oxidizable PUFA-PLs (2nd paragraph); (ii) describe key molecular characteristics of CCA subtypes and explain how these features may influence ferroptosis regulation (3rd paragraph); and (iii) summarize prior studies investigating ferroptosis regulators in iCCA or in both iCCA and eCCA models (4th paragraph). These revisions clarify the biological context and improve the rationale for investigating subtype-specific ferroptosis vulnerabilities in CCA.

Comment 2. Provide more details on the sample size and inclusion/exclusion criteria to fully assess the robustness of the design.

Response:

We thank the reviewer for this helpful comment. We acknowledge that placing the detailed description of sample selection and filtering criteria only in the Materials and Methods section may have caused confusion when interpreting the Results. In response, we have relocated the description of the analysis workflow (Figure 1) to the beginning of the Results section and expanded the explanations of the sample size, inclusion criteria, and exclusion steps to improve clarity and transparency (1st ~ 3rd paragraphs of the Results section).

Comment 3. It may be useful to consider including multivariate regression to control for possible confounding variables and gain a deeper understanding of the relationships between variables.

Response:

We appreciate the reviewer’s insightful suggestion. In this study, we had primarily assessed subtype-specific dependency using an absolute dependency-score cutoff and statistical comparisons between groups. Nonetheless, we fully agree that evaluating potential confounding variables through multivariate regression could strengthen the interpretation of associations in dependency analyses.

In response to the reviewer’s comment, we additionally performed Ordinary Least Squares (OLS) regression to examine whether the expression levels of ferroptosis defense enzymes or subtype-specific mutations independently predict GPX4 dependency (Results for Reviewer 1 are provided below). Although multivariate regression is conceptually valuable, these variables did not reach statistical significance even in the univariate model (Table 1 and 2 for Reviewer 1). Only subtype was a marginal predictor of GPX4 dependency (P = 0.05439), indicating that iCCA is associated with a 0.2693 increase in the dependency score (Results for Reviewer 1).

Collectively, although OLS-based regression analysis provides interpretable estimates of both the direction and magnitude of associations for variables, the small dataset limits statistical power and prevents more definitive conclusions. Because this study primarily evaluated iCCA selectivity for GPX4 dependency using two predefined criteria (as described in the 1st paragraph of the Results section), more comprehensive analyses of additional covariates would be better addressed in future studies designed with larger cohorts or integrated multi-omics datasets. In response to the reviewer’s suggestion, we have included a description of our exploratory analyses assessing FSG expression and subtype-specific mutations, as well as future plans to investigate subtype-dependent covariates, in the 4th paragraph of the Discussion section.

Results for reviewer 1:

In response to the reviewer’s comment, we performed OLS regression to examine whether the expression levels of ferroptosis defense enzymes, subtype-specific mutations or subtype independently predict GPX4 dependency. We first applied univariate OLS models to each variable and planned to retain variables showing nominal statistical association with GPX4 dependency for inclusion in a subsequent multivariate model. However, only subtype showed a marginal association with GPX4 dependency, so we did not proceed to a multivariate model. The results are summarized below.

First, we tested whether mRNA level of any single FSG ( could account for GPX4 dependency () using univariable OLS models as follows:

Table 1 showed the regression coefficients and P-value for nine FSGs, including AIFM2, SCD1, DHODH, MBOAT1/2, GCH1, MGST1, DHFR, and GPX4. None of the genes showed statistically significant associations with GPX4 dependency, suggesting that GPX4 essentiality is unlikely to be explained by the expression of any single FSG. Rather, it may reflect the integrated activity of a redundant antioxidant network and multiple levels of regulation, including translational and post-translational control.

Table 1. Univariable OLS regression of FSGs expression against GPX4 dependency

Gene

Coefficient

p-value

N

AIFM2

-0.090032

0.138129

28

SCD

-0.080452

0.167069

28

DHODH

0.000530

0.996660

28

MBOAT1

0.050568

0.348124

28

MBOAT2

0.030007

0.644570

28

GCH1

-0.046077

0.571709

28

MGST1

-0.019115

0.437479

28

DHFR

-0.039583

0.562631

28

GPX4

0.100972

0.263831

28

Second, we tested whether subtype-specific mutations could predict GPX4 dependency. Mutation data, available for 16 CCA cell lines in cBioPortal, were retrieved, encoded as binary mutation status, and analyzed using univariate OLS models as follows:

Table 2 summarizes the regression coefficients and P-values for mutations enriched in iCCA (IDH1/2, BAP1, ARID1A) and eCCA (ELF3, ERBB3). None of the mutations showed statistically significant associations with GPX4 dependency, likely due to the small number of mutated cell lines. The limited and subtype-imbalanced CCA models available in current datasets reduce the statistical power of OLS-based mutation analyses.

Table 2. Univariate OLS regression assessing the association between CCA-related gene mutations and GPX4 dependency.

Gene

Coefficient

p-value

Mutated Cell

Wild-type Cell

IDH1

0.043227

0.836041

3

13

IDH2

-

-

0

16

BAP1

-0.148550

0.669149

1

15

ARID1A

-0.357517

0.299518

1

15

ELF3

-0.357517

0.299518

1

15

ERBB3

-

-

0

16

Finally, we assessed whether binary subtype classification (iCCA = 1; eCCA = 0) could explain GPX4 dependency using a univariate OLS model:

The regression analysis yielded a coefficient of β₁ = 0.2693 with a P-value of 0.05439, suggesting that subtype was a marginal predictor of GPX4 dependency, with iCCA being associated with a 0.2693-unit higher dependency score. We additionally applied Fisher’s exact test by dichotomizing GPX4 dependency and constructing a 2×2 contingency table to compare the distribution of high versus low dependency between iCCA and eCCA subtypes. The analysis yielded a trend toward higher GPX4 dependency in iCCA, showing a large effect size (OR = 8.75), yet still only a borderline P-value (P = 0.069).

Comment 4. Detailed information on the experimental protocol could be provided, including validation of the methods used and justification for the choice of analytical tools.

Response:

We thank the reviewer for this helpful comment. We incorporated further details regarding (1) the rationale for method selection, (2) experimental protocol, and (3) methodological validation, as summarized below:

  • Rationale for method selection
  • Cell viability: Cell viability was assessed at 72 h after treatment using the ATP-based CellTiter-Glo® luminescence assay (Promega, Madison, WI, USA), a widely validated method known for its high sensitivity and reproducibility in drug-response studies (Section 4.3)
  • Lipid peroxidation: Lipid peroxidation was detected using the Image-iT™ Lipid Peroxidation Kit with the lipophilic BODIPY® 581⁄591 C11 probe (Thermo Fisher Scientific), which reflects lipid ROS levels through an oxidation-dependent fluorescence shift (Section 4.4)
  • Western blot analysis: To elucidate the differential sensitivities of these cell models to RSL3, we assessed the basal protein levels of GPX4, as the cellular response to a small-molecule inhibitor often depends on the abundance of its target protein. Measuring GPX4 protein levels is more informative than mRNA levels because GPX4 is a selenoprotein whose translation efficiency is tightly regulated by selenium availability (Section 2.2)
  • Experimental protocol
  • Although basic protocols for Cell viability, lipid peroxidation, and Western blot analysis were already described in Materials and Methods, we additionally provided experiment-specific conditions in the relevant parts of the manuscript.
  • Cell viability: Exposure of RSL3 for three days induced pronounced growth inhibition of iCCA cells, whereas eCCA cells displayed markedly reduced sensitivity (Section 2.2)
  • Cell viability: TFK1 cells were exposed to 5 μM XAV-939, 12.5 μM Y39983, or 12.5 μM SP600125 for 1 h, and further incubated with 1 or 10 μM RSL3 for 72 h (Section 2.3)
  • Lipid peroxidation and Western blot analysis: For panels (A–D) and (F left), lipid peroxidation was assessed by flow cytometry with BODIPY® C11 probe. For panels (E) and (F right), GPX4 expressions were analyzed by Western blotting and quantified relative to either α-tubulin or β-actin (Figure 3 Legend)

  • Validation of assay performance
  • Cell viability assay: Validation was performed using multiple compounds (RSL3, gemcitabine, doxorubicin) in independently repeated experiments, each in triplicate, yielding reproducible results (Figure 1 legend).
  • Lipid peroxidation assay: CHP (a positive control peroxide donor) increased BODIPY® C11 fluorescence (488/525 nm), confirming assay responsiveness (Figure 3D). Pre-treatment with ferrostatin-1 or deferoxamine attenuated the signal, supporting assay specificity for lipid ROS (Figure 3C).
  • Western blot analysis: GPX4 protein detection and associated assays were validated in our previous publication [47], which we have now cited in the revised manuscript (Section 4.5).

Comment 5. The discussion could address the limitations of the study and suggest future directions for research, which would strengthen the discussion and validity of the conclusions.

Response:

We appreciate the reviewer’s thoughtful comment. In response, we have added a dedicated 6th paragraph of the Discussion section that outlines the limitations of our study and proposes directions for future research. This newly added section discusses (i) the mismatch between clinically relevant CCA subtypes and available cell-line annotations, (ii) the intrinsic constraints of CRISPR-based dependency screening in capturing the complex tumor microenvironment and non-essential therapeutic targets, and (iii) the limited number of available CCA cell lines, which restricts the generalizability of our findings. We also note that, despite these limitations, the DepMap dataset remains the most comprehensive resource currently available, and the subtype-selective candidates identified in our analyses provide a basis for future mechanistic studies and preclinical validation. We believe that this expanded discussion strengthens the contextualization and interpretability of our conclusions.

 We are grateful to the reviewer for constructive input. These revisions substantially improve our manuscript’s robustness, clarity, and translational impact.

Reviewer 2 Report

Comments and Suggestions for Authors

This paper must undergo a major revision before publication

  1. In the keywords, include enzyme-inhibited ferroptosis.
  2. The introduction must be boosted with a comparative discussion on diverse enzymes (e.g: FSP1, AIFM2, DHODH, GCH1, DHFR, MGST1, SCD1, and MBOAT1/2) that participate in the inhibition of ferroptosis.
  3. The discussion section must be boosted with additional information concerning the results. Currently, this falls apart and is not up to publication standards.
  4. Equations in the materials and methods require proper abbreviation for each symbol used.
  5. In-depth discussion for statistical analyses is missing. 
  6. Provide certain concluding remarks with merits, limitations, and future direction.
  7. The resolution for Figures 1, 2, and 4 must be improved. Especially text inside the Figures looks unreadable.
  8. The reference section must be boosted with additional and recent relevant literature.
Comments on the Quality of English Language

Moderate English correction, grammatical check and typo correction is required

Author Response

Reviewer #2

December 8, 2025

Dear Reviewer #2,

We sincerely appreciate the reviewer’ time, expertise, and thoughtful critiques of our manuscript. Their comments have significantly improved the clarity, rigor, and overall scientific impact of the study. We have carefully addressed every point raised and substantially revised the manuscript accordingly. All modifications are clearly indicated in the revised version.

Comment 1. In the keywords, include enzyme-inhibited ferroptosis.

Response:

We appreciate the reviewer’s suggestion. The keyword “enzyme-inhibited ferroptosis” has been added accordingly.

Comment 2. The introduction must be boosted with a comparative discussion on diverse enzymes (e.g: FSP1, AIFM2, DHODH, GCH1, DHFR, MGST1, SCD1, and MBOAT1/2) that participate in the inhibition of ferroptosis.

Response:

We thank the reviewer for this valuable suggestion. In response, we have expanded and reorganized the 1st and 2nd paragraph of Introduction section to provide a more comparative overview of ferroptosis-regulating enzymes. Specifically, we now describe (i) the pro-ferroptotic metabolic processes that generate oxidizable PUFA-containing phospholipids—ACSL4, LPCAT3, and LOX—and (ii) the major ferroptosis defense systems that detoxify lipid peroxides—FSP1/AIFM2, DHODH, and GCH1/DHFR—or limit the availability of peroxidation-prone PUFA-PLs—SCD1, MBOAT1/2, and MGST1. These revisions clarify the mechanistic roles and distinctions among diverse ferroptosis suppressor pathways and provide a stronger rationale for investigating subtype-specific ferroptosis vulnerabilities in CCA.

Comment 3. The discussion section must be boosted with additional information concerning the results. Currently, this falls apart and is not up to publication standards.

Response:

We appreciate the reviewer’s valuable comment. In the revised manuscript, we have extensively strengthened and reorganized the Discussion section by 1) more clearly highlighting the novelty of our study in addressing the limitations of currently available CCA cell models (1st paragraph), 2) clarifying the criteria used for prioritizing FSGs and refining the mechanistic interpretation of candidate genes involved in ferroptosis regulation (2nd and 3rd paragraphs), 3) discussing preliminary analyses on the association between subtype-associated genetic alterations and GPX4 dependency (4th paragraph), 4) incorporating additional functional evidence from lipid peroxidation assays supporting the combinatorial effects of WNT inhibition and GPX4 blockade (5th paragraph), and 5) adding a dedicated paragraph that outlines key study limitations and future research directions (6th paragraph). Collectively, these revisions improve the logical coherence of the Discussion and more effectively integrate our findings with underlying biological mechanisms and previously reported literature.

Comment 4. Equations in the materials and methods require proper abbreviation for each symbol used.

Response:

We thank the reviewer for pointing this out. In accordance with the comment, we revised the Materials and Methods (Section 4.2) to provide explicit definitions of all symbols used in the equations.

Comment 5. In-depth discussion for statistical analyses is missing. 

Response:

We appreciate the reviewer for helpful comment. In accordance with the suggestion, we revised the Results, Figure legend and Materials and Methods section to clarify the explanations about the rationale for choosing specific statistic testing. In addition, we updated the Figure Legends to clearly indicate the statistical methods applied. Revisions were incorporated in the following sections:

  • Results (Section 2.1)

Dependency scores were compared using the Wilcoxon rank-sum test, given the non-normal distribution of the data and the sample-size imbalance. To quantify the direction and magnitude of subtype-selective effects, we calculated Hedges’ g, which provides a standardized effect size corrected for small-sample bias [21]. The results were visualized in a volcano plot, with Hedges’ g on the x-axis and −log10(P-value) on the y-axis (Figure 1C)

  • Figure Legends (Figures 3, 4, and 5)

Data are presented as mean ± standard deviations (SD) from 3–4 independent experiments, each performed in triplicate. Statistical significance was determined by one-way ANOVA followed by Tukey’s post hoc test.

  • Materials and Methods (Section 4.7)

Depending on data distribution and the type of comparison type, statistical analyses were performed using one-way ANOVA with Tukey’s post hoc test for multi-group comparisons and Wilcoxon rank-sum test for two-group comparisons

Comment 6. Provide certain concluding remarks with merits, limitations, and future direction.

Response: We appreciate the reviewer’s thoughtful suggestion. In accordance with the comment, we revised the last paragraph of Discussion section to more clearly articulate the merits, current limitations, and future directions of our study. This revised conclusion provides a more comprehensive and forward-looking summary that aligns with the reviewer’s recommendation.

Comment 7. The resolution for Figures 1, 2, and 4 must be improved. Especially text inside the Figures looks unreadable.

Response: We appreciate the reviewer’s comment. As requested, Figures 1, 2, and 4 have been regenerated at high resolution. The text and embedded annotations were fully revised to improve readability, and color schemes were standardized for visual consistency across figures. In addition, all figure panels were exported directly from the original analysis scripts to avoid rasterization artifacts. We believe these updates substantially enhance the clarity and presentation quality of the figures.

Comment 8. The reference section must be boosted with additional and recent relevant literature.

Response:

We appreciate the reviewer’s suggestion. In the revised manuscript, we expanded the reference list from 21 to 47 entries by incorporating additional and more recent literature relevant to ferroptosis biology and cholangiocarcinoma research. These updates ensure that the Introduction and Discussion are better grounded in current scientific advancements and provide appropriate context for our findings.

We are grateful to the reviewer for constructive input. These revisions substantially improve our manuscript’s robustness, clarity, and translational impact.

Best regards,

Seung Jin, Lee, Ph.D.

Round 2

Reviewer 2 Report

Comments and Suggestions for Authors

I appreciate the authors' effort in improving the manuscript. I recommend its publication